# A Size-Controlled AFGAN Model for Ship Acoustic Fault Expansion

**Linke Zhang** [1,2,†]**, Na Wei** [3,*,†]**, Xuhao Du** [4,*] **and Shuping Wang** [3]

1   Key Laboratory of High Performance Ship Technology, Wuhan University of Technology, Ministry of Education, Wuhan 430063, China; lincol@whut.edu.cn
2   Key Laboratory of Marine Power Engineering & Technology, Wuhan University of Technology, Wuhan 430063, China
3   Key Laboratory of Modern Acoustics and Institute of Acoustics, Nanjing University, Nanjing 210000, China; wangsp822105@126.com
4   Department of Mechanical Engineering, The University of Western Australia, Crawley 6008, Australia
*   Correspondence: weina1223@126.com (N.W.); xuhao.du@uwa.edu.au (X.D.)
†   These authors contributed equally to this work.

**Abstract:** Identifying changes in the properties of acoustical sources based on a small number of sample data from measurements has been a challenge for decades. Typical problems are the increasing sound power from a vibrating source, decreasing transmission loss of a structure, and decreasing insertion loss of vibration mounts. Limited access to structural and acoustical data from complex acoustical systems makes it challenging to extract complete information of the system and, in practice, often only a small amount of test data is available for detecting changes. Although sample expansion via interpolation can be implemented based on the priori knowledge of the system, the size of the expanded samples also affects identification performance. In this paper, a generative adversarial network (GAN) is employed to expand the acoustic fault vibration signals, and an Acoustic Fault Generative Adversarial Network (AFGAN) model is proposed. Moreover, a size-controlled AFGAN is designed, which includes two sub-models: the generator sub-model generates expanded samples and also determines the optimal sample size based on the information entropy equivalence principle, while the discriminator sub-model outputs the probabilities of the input samples belonging to the real samples and provides the generator with information to guide sample size considerations. Some real data experiments have been conducted to verify the effectiveness of this method.

**Keywords:** acoustic fault identification; sample expansion; generative adversarial network; size control

## 1. Introduction

Monitoring the condition of the ship structure has been a challenge for decades [1,2]. Two general approaches are considered including the theoretical and numerical simulation method and the experimental method. Due to the complexity of the ship structure and its components inside, accurate simulation of its dynamic property is difficult and time-consuming [1,3]. Thus, researchers try to establish the condition monitoring model based on actual experimental data [4,5] and among all other methods, using the vibration and acoustic data provide a safe, non-invasive, and constant condition monitoring [2]. Li et al. in 2012 proposed a support vector machine model to detect the oil leakage condition of the ship. His model was built based on 300 samples of instantaneous angular speed, which achieved 94.7% accuracy [5]. He also combines independent component analysis, principal component analysis to model to the normal and fault engine conditions based on 100 samples each [6]. This model presents 90.5% accuracy.

However, as shown by previous studies, practical acoustic fault identification in ships is considered as a small-sample recognition problem because of the difficulty of obtaining representative fault samples, the high experimentation cost, and so on [2]. Typical problems are increasing sound power from a vibrating source, decreasing transmission loss of the structure, and decreasing insertion loss of vibration mounts. Sample expansion is an effective way to solve the problem of incomplete samples in ship acoustic fault source identification [2]. In general, there are two main methods for constructing expanded samples. One is to obtain expanded samples by introducing the prior knowledge, which can be regarded as an effective supplement to actual fault samples. Niyogi et al. proposed a virtual sample generation technology based on geometric transformation in an image recognition field [7]. For noise source recognition in a double-layer cylindrical shell structure, Xu et al. generated virtual samples based on a system frequency response function, which demanded a deep understanding of a priori knowledge from field researchers [8]. The other method is to expand samples by employing the concept of a disturbance, such as bootstrapping [9], noise disturbance [10], and so on. The advantage of such methods is that the expanded samples can be constructed directly from the original training samples. However, bootstrapping constrains itself by picking up the same sample multiple times, which introduces no new information. Furthermore, signal augmentation like adding noise disturbance might generate poor samples due to the ignorance of the field's priori knowledge. Other common sample expansion methods like interpolation and synthetic minority over-sampling technique can present good performance when the sample is represented by a set of features, which are not applicable to time series data like acoustic data. Therefore, the lack of priori knowledge and the nature of acoustic data make the sample expansion difficult.

The generative adversarial network (GAN) provides a method to directly generate the acoustic signal under a fault condition. In 2014, Goodfellow et al. proposed the GAN and successfully applied it to the field of computer vision by generating a large number of highly realistic images [11]. Instead of assuming the distribution of functions first, the real data were directly approximated by sampling in the GAN. This anti-competitive approach ensured the quality of the generated samples.

The GAN approach has been applied to image recognition [12], audio [13], video [14], and other fields [15–18]. However, only a small amount of work has been reported related to fault vibration signal identification [19]. This paper attempts to apply a GAN to acoustic fault sample expansion under small-sample conditions and presents an Acoustic Fault Generative Adversarial Network (AFGAN) model, which can directly expand the acoustic sample under fault condition. Furthermore, a size-controlled AFGAN is designed to solve the "information hedge" problem in acoustic source identification by constructing the generator according to the information entropy equivalence principle of the expanded samples and the original ones.

The rest of this paper is organized as follows. In Section 2, we discuss the GAN for acoustic fault sample expansion, including the GAN setup (Section 2.1) and the proposed AFGAN model details (Section 2.2). A size-controlled AFGAN is designed in Section 3, where a sample size control function is derived based on the principle of information entropy equivalence (Section 3.1), and the resulting objective function for size-controlled AFGAN is presented (Section 3.2). Real data studies are conducted in Section 4. Some concluding remarks are given in Section 5.

## 2. GAN for Acoustic Fault Sample Expansion

### 2.1. GAN

The basic concept of the GAN is to learn the potential distribution of real samples from training samples through adversarial learning, and then generate a large number of training samples. It includes two components: the generator and the discriminator. The generator is used to capture the potential distribution of real samples and generate expanded samples (i.e., fake samples). The discriminator is a binary classifier and predicts whether the input is a real sample or a fake sample. The GAN trains with the adversarial learning process: the generator generates fake samples that look like real samples to

cause the discriminator to misjudge, and the discriminator should identify the fake samples as often as possible.

The performance of the GAN entirely depends on the network structure of the generator and discriminator. Early GANs adopted fully connected layers and maxout output layers [11]. The Deep Convolutional GAN (DCGAN) [20] combines a GAN with a convolutional neural network, and adopts fractional-strided convolutions and strided convolutions to generate the images, which makes the GAN training process more stable and also generates higher-quality samplings.

### 2.2. AFGAN Network Architecture

The AFGAN is modelled by employing the DCGAN structure. Unlike a conventional GAN which generates two-dimensional images, the AFGAN generates one-dimensional signals. A schematic diagram of the AFGAN architecture is shown in Figure 1.

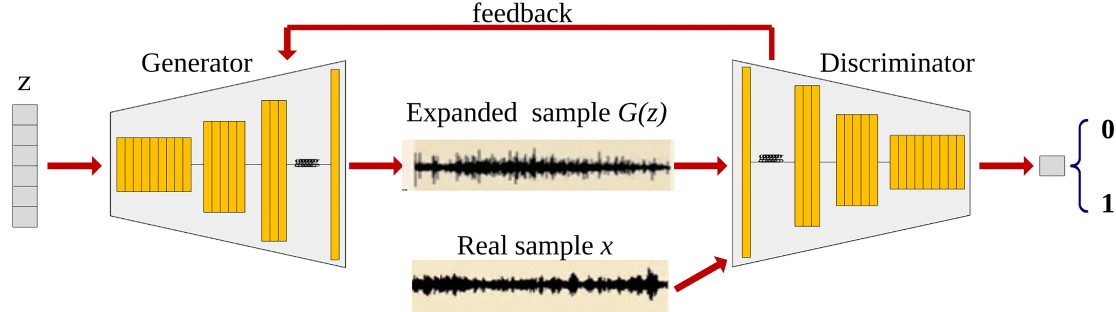

**Figure 1.** The Acoustic Fault Generative Adversarial Network (AFGAN) architecture.

The generator performs signal expansion. Its inputs are the latent representation $z$ ($z \in \mathbb{R}^Q$), and its output is the expanded signal $G(z)$ ($G(z) \in \mathbb{R}^p$). The generator is designed to be fully convolutional, which makes the model focus only on the time correlation of the input signal. Its input is uniformly distributed noise $z$, and then its output is reshaped into a four-dimensional tensor. Finally, a single-channel acoustic fault signal $G(z)$ can be generated through multiple fractional-strided convolution layers. The output layer adopts the Tanh activation function while the other layers adopt the ReLU activation function [20]. For the discriminator, the input signal, i.e., expanded signal $G(z)$ or the real signal $x$, is converted to a single-feature map through multiple convolution layers in order to tell whether it comes from real or generated samples. The leaky ReLUn [20] is adopted in the activation function in all the layers.

An important feature of the AFGAN model is its end-to-end structure. The original signal does not need to be specially processed, such as undergoing feature extraction and can be directly inputted into the AFGAN model to generate expanded samples.

## 3. The Size-Controlled AFGAN

It is useful to improve the fault recognition rate by using a GAN to expand acoustic fault samples under small-sample conditions. However, an increased number of generated samples does not necessarily lead to better results. Too much expanded sample information may overwhelm real sample information and results in the "information hedge" problem, which may decline the recognition performance. Therefore, it is necessary to control the expanded sample size.

### 3.1. The Information Entropy Equivalence Principle

The GAN model generates expanded samples by capturing the potential distribution of real samples. From the point of information theory, the information contained in the expanded samples should be equal to the information provided by the real samples. The "information hedge" problem

during the acoustic fault sample expansion process can be explained as an issue that the amount of the information of the expanded samples is more or less than that of the real samples, which leads to the distortion of expanded samples. Therefore, to avoid the "information hedge", it is necessary to ensure that the same amount of information is contained in both two sample sets, and thus the size of the expanded samples can be optimized.

Information entropy is employed to measure the amount of information contained in the fault samples. Assume the real sample set and the generated sample set are $X = x_1, x_2, ..., x_M$ and $gX = G(z_1), G(z_2), ..., G(z_N)$, respectively, where $x_i \in \mathbb{R}^p$, $z_i \in \mathbb{R}^Q$. The discriminator outputs the probabilities of the input samples belonging to the real samples. For sets $X$ and $gX$, their outputs after the discriminator are $D(x_i)$ and $D(G(z_j))$, respectively.

For the real samples, the information entropy [21] of the sample $x_i$ can be calculated with

$$Hx_i = -D(x_i)logD(x_i) - (1 - D(x_i))log(1 - D(x_i)). \tag{1}$$

Thus, the amount of information of the real samples set $X$ is

$$\begin{aligned} HX &= \frac{1}{N}\sum_{i=1}^{M} Hx_i \\ &= -\frac{1}{M}\sum_{i=1}^{M}[D(x_i)logD(x_i) + (1 - D(x_i))log(1 - D(x_i))]. \end{aligned} \tag{2}$$

Equation (2) denotes the average information entropy of the real sample set.

For the expanded sample set, the samples are rearranged in descending order of $D(G(z_j))$, and the rearranged samples are $gX_{sort} = G(z_{i1}), G(z_{i2}), ..., G(z_{iN})$, where $1 \geq D(G(z_{i1})) \geq D(G(z_{i2})) \geq ... \geq D(G(z_{iN})) \geq 0$. The average information entropy of the first $K$ expanded samples can be calculated with

$$\begin{aligned} H(gX(K)) &= H(gX_{sort}(K)) \\ &= -\frac{1}{K}\sum_{i=1}^{K}[D(G(z_{i_k}))logD(G(z_{i_k})) + (1 - D(G(z_{i_k})))log(1 - D(G(z_{i_k})))]. \end{aligned} \tag{3}$$

According to the principle of information entropy equivalence, we need to find an expansion subset with size $K^*$, so that the information entropy difference between the subset and real sample set reaches the minimum, which is

$$\begin{aligned} K^* &= argminL_{SIZE}(G) \\ &= \underset{K=1,...,N}{argmin}|H(X) - H(gX(K))| \\ &= \underset{K=1,...,N}{argmin}| -\frac{1}{M}\sum_{i=1}^{M}[D(x_i)logD(x_i) + (1 - D(x_i))log(1 - D(x_i))] + \\ &\quad \frac{1}{K}\sum_{i=1}^{K}[D(G(z_{i_k}))logD(G(z_{i_k})) + (1 - D(G(z_{i_k})))log(1 - D(G(z_{i_k})))]|. \end{aligned} \tag{4}$$

Equation (4) shows that a smaller $L_{SIZE}(G)$ implies that the two information entropy is more likely to be equivalent. In other words, the smaller the $L_{SIZE}(G)$, the more equal the information amount of the subset and real sample set, which is useful for the sample size control.

*3.2. The Generator Objective Function*

The objective function of the traditional GAN is

$$L_{GAN}(D,G) = E_{x \sim p_{data}(x)}[logD(x)] + E_{z \sim p_z(z)}[log(1 - D(G(z)))], \tag{5}$$

where $E[\ ]$ is the mathematical expectation, and $P_{data}()$ and $P_z()$ are the probability density distributions of the real samples and random noise, respectively. In the GAN training process, the generator adjusts parameters to minimize $L_{GAN}(D,G)$, while the discriminator adjusts parameters to maximize $L_{GAN}(D,G)$. With a number of iterations, the final objective is

$$G* = argmin_G max_D L_{GAN}(D,G). \tag{6}$$

If the size constraint is taken into consideration, the task of the discriminator remains the same, which is to calculate the probabilities that the generated samples belong to the real samples. In addition to generating samples, the generator also needs to control the sample size. The objective function can be expressed as

$$G* = argmin_G max_D(1 - \lambda)L_{GAN}(D,G) + min_G \lambda L_{SIZE}(G), \tag{7}$$

where $\lambda$ is a constant (0 or 1). During the training period, $\lambda = 0$ and the size-controlled AFGAN function is used to train the generator and discriminator to generate realistic expanded samples. After the network has been trained stably, $\lambda = 1$, and its function is used to control the expanded sample size. The generator will continue generating samples until the size has reached the optimal value.

## 4. Experimental Setup

### 4.1. The Measured Noise Source Data Set

The measured data obtained from a 1:1 ship module model is used to validate the proposed method. The sampling frequency is 2048 Hz. The primary noise sources are the motor, the pump, and the high-frequency exciters, representing three typical fault sources with main vibration frequencies 90 Hz, 296 Hz and 360 Hz, respectively. Accelerometers PCB 352CC were placed on the above three devices to collect the vibration signal and recorded by an 8-channel B&K 3560D pulse system. In addition, 1024 sampling points in the time domain are chosen as observation samples. The positive frequency bands of the signal power spectrum are obtained as feature vectors, and its dimension is 512. There are 900 mechanical noise samples obtained for each class, from which 800 measured samples are chosen as training samples and the remaining 100 as testing samples.

### 4.2. Network Configuration

The size-controlled AFGAN model is designed based on the measured data set. The input to the generator has a uniform distribution with a 100-dimensional noise vector z, and z is projected onto a four-dimensional convolutional representation with eight fractional-strided convolution layers. The convolution kernel size is $5 \times 1$. For each layer, the convolution kernel numbers $\times$ the signal lengths are $2048 \times 8$, $1024 \times 16$, $512 \times 32$, $256 \times 64$, $128 \times 128$, $64 \times 512$, $32 \times 1024$, $16 \times 1024$, and finally $1024 \times 1$-dimensional single-channel signals are output through these characterization transformations. The output layer in the generator uses the Tanh function, and the other layers use the ReLU activation functionn [20].

The discriminator was also a fully convolutional network, building the reverse process of the generator. The input signal is a $1024 \times 1$-dimensional single-channel vector. There are eight strided convolution layers. The channel number doubles and the signal length halves (except in the second layer, whose signal length remains the same as the first layer) per layer compared with the previous layer. The final output is a scalar, which denotes the probability of the generated sample belonging to the real training data. The Leaky ReLU activation functionn [20] is used in all convolution layers in the discriminator.

*4.3. Sample Expansion*

　　To ensure the training stability, the input signals are pre-processed to zero-mean and normalized to [−1, 1]. All the weights of the network are initialized from a zero-mean normal distribution with a standard deviation of 0.02. In the Leaky ReLU, the slope of the leak is set to 0.2 in all models. The hyperparameter is adjusted using Adam optimization, and the learning rate is 0.0002. In addition, 800 real samples are trained with the size-controlled AFGAN for sample expansion. Figure 2 shows the frequency domain plots of both the real samples (Figure 2a–c) and the generated samples (Figure 2d–f). It can be seen that the main vibration frequency of the expanded samples is quite consistent with that of the real samples.

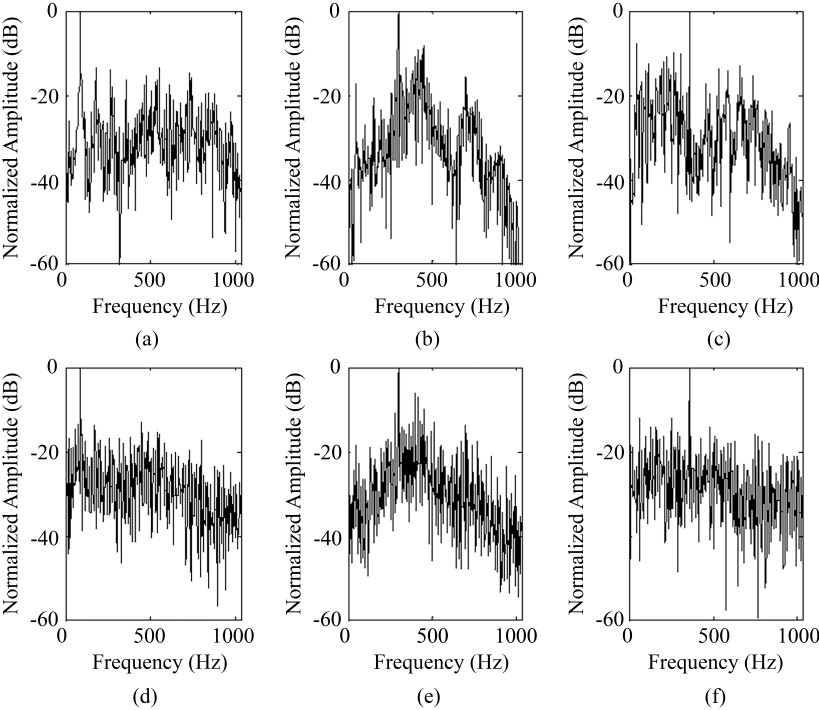

**Figure 2.** Frequency domain plots of the measured signals with main vibration frequencies (**a**) 90 Hz; (**b**) 296 Hz; (**c**) 360 Hz and the generated signals with main vibration frequencies; (**d**) 90 Hz; (**e**) 296 Hz; (**f**) 360 Hz.

*4.4. Recognition Accuracy with Different Expanded Sample Sizes*

　　The generated samples and the real samples are further combined into a new training sample set and then applied to acoustic fault identification. A Multi-layer Perceptron (MLP) artificial neural network [22] is employed as the classifier. The classifier performance of the proposed algorithm is tested on the optimum sample size and also compared with different expanded sample sizes. The total number of generated samples is 1000 for each class. The expanded training dataset consists of two parts, 800 real samples and some expanded samples, which have been rearranged according to the descending order of their confidences. Based on Equation (7), the optimum sample size for three classes are $n_1 = 831$, $n_2 = 538$, $n_3 = 282$ respectively. Design contrast experiments as: $E_1$: the optimum size; $E_2$: half the optimum size; $E_3$: double the optimum size, if the size is larger than 1000, then 1000 is chosen; $E_4$: all 1000 samples; $E_5$: none. The classifiers' accuracies of these different expanded sample sizes are compared and listed in Table 1.

**Table 1.** Accuracies of different expanded sample sizes.

| Experiment | Sample Size per Typical Fault Source | | | Accuracy |
| --- | --- | --- | --- | --- |
| | **90 Hz** | **296 Hz** | **360 Hz** | |
| $E_1$: the optimum size | 831 | 538 | 282 | 83.00% |
| $E_2$: half the optimum size | 416 | 269 | 141 | 82.67% |
| $E_3$: double the optimum size | 1000 | 1000 | 564 | 81.67% |
| $E_4$: all 1000 samples | 1000 | 1000 | 1000 | 83.00% |
| $E_5$: none | 0 | 0 | 0 | 61.76% |

In Table 1, the accuracies (83.00%, 82.67%, 81.67%, 83.00%) for the first four experiments are significantly higher than the accuracy with no expanded samples addition (61.76%). This shows that the AFGAN for sample expansion is effective. At the optimum sample size, the accuracy reaches the highest 83.00%, half or double the optimum sample size leads to the decrease of the accuracy. When the sample size is chosen 1000 per class, the model's accuracy also reaches 83.00%, which is the same as that of the optimum size. This shows that all those generated quality samples are capable of providing a high identification performance, but there is a trade-off between the number of samples and the time for classification. Thus, sample size control is necessary.

Furthermore, the relationships among the classier performance, the expanded sample arrangement and the sample size are investigated. Three expanded sample arrangement schemes, random, descending and ascending order are carried out. Figure 3 illustrates the classifier accuracies as the sample size variation on these three schemes. It is clear that the accuracy of the descending order outperforms the ascending one. To reach the accuracy 83.00%, which is obtained based on the optimum size, the n1, n2 and n3 need to be 500 in the descending order scheme, while the ascending one needs all the 3000 samples. This behaviour demonstrates that the descending order rearrangement can ensure the high confidence samples be selected preferentially so that the high accuracy can be achieved through small sample size. It is also worth pointing out that, for the descending order scheme in sample size range [850, 900], the accuracy reaches the 83.33%, which is a little higher than the accuracy when the sample size is optimal, in the sense that an additional increase in sample size does not lead to further sensible improvements in classifier performance. Thus, the results demonstrate that the performance of the size-controlled AFGAN can be considered satisfactory with the optimum sample size.

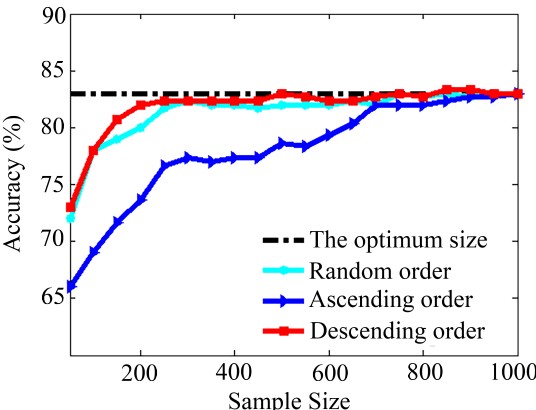

**Figure 3.** The accuracies as the sample size variation for different arrangement schemes.

## 4.5. Sample Expansion Performance on Other Machine Learning Algorithms

To test the generalization of the expanded samples generated by the size-controlled AFGAN, some models are built on different machine learning algorithms, and their classification performance before and after expansion at the optimum sample size are compared. The algorithms used in this task

contain the most common machine learning algorithm like MLP neural network, passive aggressive classifier (PAC) [23], ridge classifier [24], extreme gradient boosting classifier (Xgboost) [25], random forest (RF) [26], and gradient boosting decision tree (GBDT) [27].

To eliminate the uncertainty, 11 different seeds (0, 50, ..., 500) are set for the different learning process to obtain different models. Therefore, there are a total of 46 models developed for validating the generalization of the expanded data, where the data shuffle process does not influence Xgboost and ridge classifier. To demonstrate the performances' differences based on data before and after expansion, the absolute accuracy increase (AAI) and relative error reduction (RER) are calculated. RER is used to eliminate the original performance of the model, which means when the original model has already achieved high accuracy, the AAI must be smaller than the one with lower accuracy, so the RER is calculated to eliminate this influence. RER is calculated with

$$RER = \frac{Acc_{ae} - Acc_{be}}{Acc_{be}}. \tag{8}$$

Figure 4a presents the AAIs of different models versus different algorithms and Figure 4b presents the RERs. Different colours represent different shuffle seeds. The average AAI and RER are 11.6% and 33.0%, and there are 93.5% (i.e., 43/46) models improved their accuracies. Table 2 summarized the average performances of the models from each algorithm. Most of the models have improved their performance after sample expansion.

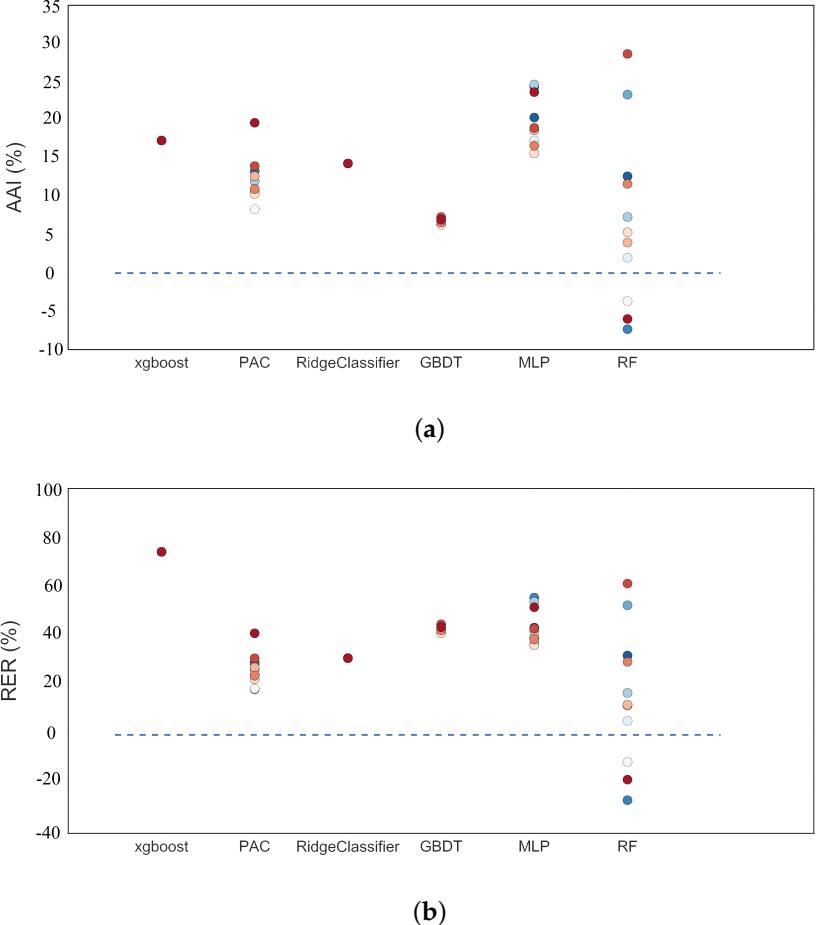

(**a**)

(**b**)

**Figure 4.** The accuracies (**a**) increase and (**b**) relatively error reduction under different machine learning algorithms and shuffle seeds.

**Table 2.** Average performances of the models from each algorithm.

| Algorithm | Mean Absolute Accuracy Increase | Improved Models Percentage |
|---|---|---|
| MLP | 19.4% (43.8%) | 100% |
| Passive Aggressive Classifier | 12.2% (26.6%) | 100% |
| Ridge Classifier | 14.3% (31.2%) | 100% |
| Extreme Gradient Boosting Classifier | 17.3% (74.3%) | 100% |
| Random Forest | 7.1% (15.2%) | 72.7% |
| Gradient Boosting Decision Tree | 6.8% (42.8%) | 100% |

To present the details of the models, the recognition accuracies of 11 GBDT models and MLP before and after sample expansion are given in Figure 5. The average AAI of GBDT before expansion is 84.2%, and, after using the expanded samples, the average recognition accuracy increases to 91.0%. As for the models built on MLP, average accuracy before expansion is 55.7%, after using the expanded samples, the average recognition accuracy increases to 75.1%. For these two nonlinear algorithms, the performances have obtained significant improvement even if the parameters were not specially tuned. The best model even achieved 92% detection accuracy, which is comparable with the previous studies even with different fault conditions [5,6].

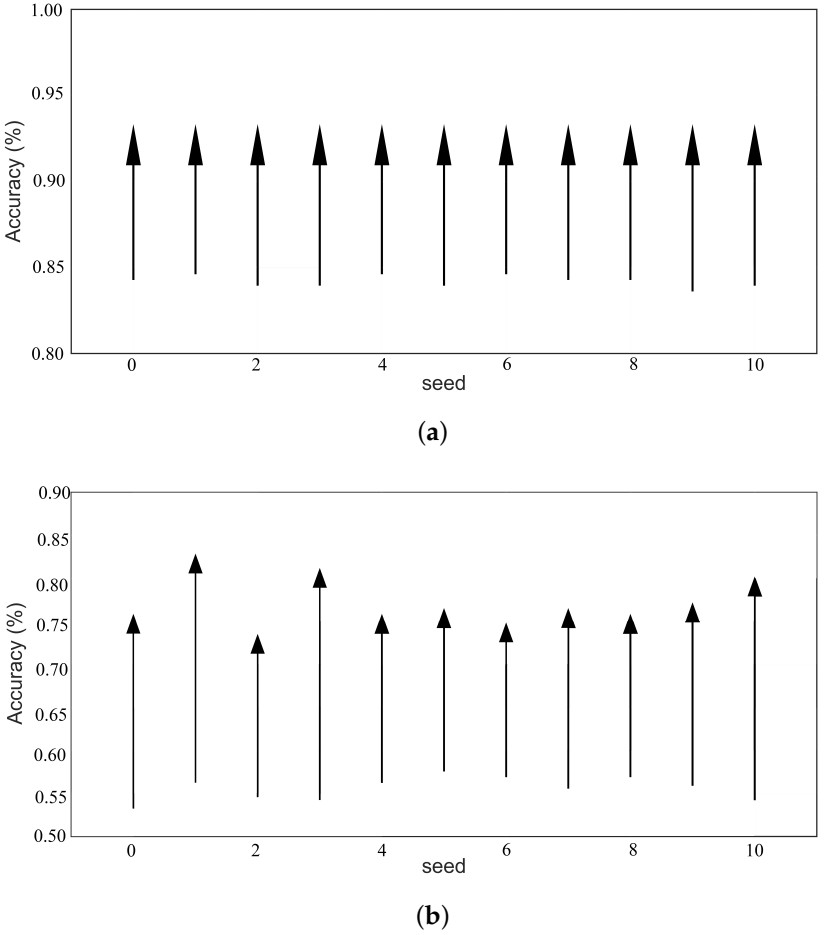

**Figure 5.** The performance increase after sample expansion of 11 (**a**) Gradient Boosting Decision Tree and (**b**) Multi-Layer Perceptron models.

The models with which RER improved the most are the GBDT, Xgboost and MLP classifier, which are better in discovering the nonlinear structure of the data compared to other algorithms. Since the expanded samples were generated by two convolutional neural networks, the expanded samples must have a more nonlinear relationship with the original samples. Therefore, it is reasonable that models built on algorithms for nonlinear structure tend to show more significant improvement. From Figure 4b, a decrease in the performance of three models has been observed because they fall into the local minimum dip during the training caused by the data shuffle. As can be seen in Figure 4a,b, different data shuffle has a significant impact on the models built on the RF algorithm, three of the RF models' performances decrease. However, the decrease occurs only 6.5% with average −6% AAI and −20% RER, which is small compared to the increase of other models. The generalization ability to use size-controlled AFGAN to generate new samples for building a more robust model has been verified with this experiment at a specific extension.

## 5. Conclusions

In this paper, a GAN was applied to the expansion of acoustic fault samples under small-sample conditions. The original signal does not need to be specially processed, such as undergoing feature extraction and can be directly inputted into the AFGAN model to generate expanded samples. Then, the information entropy equivalence principle was employed in the AFGAN model for sample size control while producing high-quality expanded samples. The proposed sample expansion method is more intuitive to the nature of the problem compared to other traditional sample expansion algorithms. In the application of mechanical noise source recognition, the results showed that the samples obtained by the size-controlled AFGAN had the advantages of high quality and optimal size, and the classifier performance was optimal with the optimal sample size. The generalization was proved by applying the expanded data to other typical machine learning algorithms. Forty-six models were trained and an 11.6% average accuracy increase and 33.0% relative error reduction were achieved.

**Author Contributions:** Conceptualization, L.Z. and X.D.; methodology, L.Z., N.W. and X.D.; software, L.Z., N.W. and X.D.; validation, L.Z., N.W., X.D. and S.W.; formal analysis, L.Z., N.W. and X.D.; investigation, L.Z., N.W. and X.D.; resources, L.Z., N.W. and X.D.; data curation, L.Z. and N.W.; writing—original draft preparation, N.W.; writing—review and editing, X.D. and S.W.; visualization, L.Z., N.W. and X.D.; supervision, L.Z.; project administration, L.Z.; funding acquisition, L.Z.

**Funding:** This research was funded by the National Natural Science Foundation of China under Grant No. 51205404 and 51709216.

**Acknowledgments:** This work was supported by the National Natural Science Foundation of China under Grant Nos. 51205404 and 51709216.

**Conflicts of Interest:** The authors declare no conflict of interest.

## Abbreviations

The following abbreviations are used in this manuscript:

| | |
|---|---|
| GAN | Generative Adversarial Network |
| AFGAN | Acoustic Fault Generative Adversarial Network |
| DCGAN | Deep Convolutional Generative Adversarial Network |
| MLP | Multi-Layer Perceptron |
| PAC | Passive Aggressive Classifier |
| Xgboost | Extreme Gradient Boosting Classifier |
| GBDT | Gradient Boosting Decision Tree |
| RF | Random Forest |
| AAI | Absolute Accuracy Increase |
| RER | Relative Error Reduction |

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
