# Peer review of "A Size-Controlled AFGAN Model for Ship Acoustic Fault Expansion"

_applsci, doi:10.3390/app9112292_

Round 1
Reviewer 1 Report
Manuscript applsci-488163-peer-review-v1
In the reviewer's opinion:
* The manuscript presents interesting research
* In the sections reference and introduction authors must extended discussion of research results of others scientists
* section 4 - too little information about the measurement method and significant measurement assumptions, but also: sampling frequency, signal lengths, etc.
* lack a discussion of research results in relation to the state of knowledge
In the reviewer's opinion, this manuscript after major improvement, should be presented in Applied Sciences.
Regards
Author Response
* In the sections reference and introduction authors must extended discussion of research results of others scientists
Response and action: Thank you for your suggestion! We have added more discussion in the introduction section (Line 19 - 29, Line 44 - 50).
* section 4 - too little information about the measurement method and significant measurement assumptions, but also: sampling frequency, signal lengths, etc.
Response and action: Thank you for your suggestion! We have included more information about the experimental setup (Line 151 - 155).
* lack a discussion of research results in relation to the state of knowledge
Response and action: Thank you for your suggestion! We have added more discussion in relation to the state of knowledge (Line 19 - 29, Line 44 - 50, Line 244 - 246). It is a little bit difficult to compare to the existing research because the fault conditions we have are occur on the pump, motor, and high-frequency exciter while other studies are not and their technique is not easy to apply directly. While the proposed method still shows its potential by comparing the accuracy before and after data expanded. Therefore, we believe most machine learning model can benefit from our proposed AFGAN method by expanding the small fault sample size.
Reviewer 2 Report
This submission presents a new model to produce data needed for identifying changes in the properties of acoustical sources based on a small number of sample data from measurements, mainly for acoustic problems in a ship.
The methodology is well described and the results and discussion are clear. Therefore, this reviewer is positive for publication of this submission.
However, in the same time this reviewer considers that there is some weakness in clarifying and justifying the needs and accuracy of the present model. Usually, when small number of data only are available, interpolation or numerical simulations (often they are energy based, though) are employed. Therefore, the authors should, if possible, discuss and explain how the present method is superior to the existing methods, and the comparisons of the accuracy in the prediction in some acoustic criteria.
The proposal of a new method is usually required to show its merit.
This reviewer believes that it is not difficult to add some explanation sentences for the authors.
Author Response
Response and action: Thank you for your comments! We have added more discussion of our result to previous study (Line 19 - 29, Line 60, Line 244 - 246, 264 - 265). The improvement after expanded data has also been highlighted (Line 239 - 246). It is a little bit difficult to compare to the existing research because the fault conditions we have are occur on the pump, motor, and high-frequency exciter while other studies are not and their technique is not easy to apply directly. While the proposed method still shows its potential by comparing the accuracy before and after data expanded. Therefore, we believe most machine learning model can benefit from our proposed AFGAN method by expanding the small fault sample size.